# Pre-trained Vision-Language Models Learn Discoverable Visual Concepts

**Yuan Zang**                                                    *yuan_zang@brown.edu*
*Department of Computer Science*
*Brown University*

**Tian Yun**                                                     *tian_yun@brown.edu*
*Department of Computer Science*
*Brown University*

**Hao Tan**                                                      *hatan@adobe.com*
*Adobe Research*

**Trung Bui**                                                    *bui@adobe.com*
*Adobe Research*

**Chen Sun**                                                     *chensun@brown.edu*
*Department of Computer Science*
*Brown University*

**Reviewed on OpenReview:** *https://openreview.net/forum?id=Vq0wMFBjo2*

## Abstract

Do vision-language models (VLMs) pre-trained to caption an image of a *durian* learn visual concepts such as *brown* (color) and *spiky* (texture) at the same time? We aim to answer this question as visual concepts learned "for free" would enable wide applications such as neuro-symbolic reasoning or human-interpretable object classification. We assume that the visual concepts, if captured by pre-trained VLMs, can be extracted by their vision-language interface with text-based concept prompts. We observe that recent works prompting VLMs with concepts often differ in their strategies to define and evaluate the visual concepts, leading to conflicting conclusions. We propose a new concept definition strategy based on two observations: First, certain concept prompts include shortcuts that recognize correct concepts for wrong reasons; Second, multimodal information (e.g. visual discriminativeness, and textual knowledge) should be leveraged when selecting the concepts. Our proposed concept discovery and learning (CDL) framework is thus designed to identify a diverse list of generic visual concepts (e.g. *spiky* as opposed to *spiky durian*), which are ranked and selected based on visual and language mutual information. We carefully design quantitative and human evaluations of the discovered concepts on nine diverse visual recognition datasets, which confirm that pre-trained VLMs do learn visual concepts that provide accurate and thorough descriptions for the recognized objects. Code and models are publicly released. [1]

## 1 Introduction

Can a vision-language model (VLM) pre-trained on images of *California seagull* learn visual concepts (Lake et al., 2015; 2011; Lee et al., 2023; Sun et al., 2015), such as *yellow legs* and *white belly*, to describe an image of a *Gentoo penguin*, which the model might not see during training? Visual concepts such as color, shape, and texture help models generalize compositionally (Farhadi et al., 2009; Nagarajan & Grauman,

---

[1]Project page and code: https://conceptdiscovery.github.io

**California Seagull**

| In VDES, CLIP aligns the image with concepts: | After removing category names, CLIP aligns the image with concepts: | After randomly combine category names with concepts, CLIP aligns the image with: | With CDL, CLIP aligns the image with: |
|---|---|---|---|
| California Gull, which has a yellow ring around its black eyes | fish-eating bird | California Gull, which has a black cap with white feather streaks | yellow legs |
| California Gull, which is a fish-eating bird | long, narrow wings | California Gull, which is a colorful bird | black, tapered tail |
| California Gull, which has a long, black tail | a nocturnal bird of prey | California Gull, which has yellow eyes | white belly |
| California Gull, which has gray back and wings | gray belly | California Gull, which is four-limbed | a white bird with a short neck |

Figure 1: The top-selected text prompts by CLIP given a query image, VDES is proposed by Menon & Vondrick (2022), whereas CDL is our proposed approach. The design of concept prompts plays a critical role on understanding whether VLMs learn visual concepts. We can observe that concept-augmented prompt can predict correct visual concepts (e.g., *gray back and wings*) when the prompt is associated with the category name (*California seagull*). When the category name is removed from the prompt (Column 2), the retrieved concepts are either non-visual or incorrect. We attribute this to the category name bias (Column 3), as the correct category can be retrieved by CLIP even when the paired descriptions are randomly shuffled and thus irrelevant. We propose a concept discovery and learning (CDL) framework and demonstrate that pre-trained VLMs can indeed learn visual concepts (e.g., Column 4). Correctly predicted concepts are in green, wrong concepts are in red, and non-visual concepts are in violet. Category names are in orange.

2018; Hsieh et al., 2024; Thrush et al., 2022; Stein et al., 2024), and can be incorporated into neuro-symbolic frameworks (Mao et al., 2019; Yi et al., 2018) or offer concept-based explanations for classification decisions (Koh et al., 2020).

Our paper aims to investigate if VLMs, such as CLIP (Radford et al., 2021), learn visual concepts automatically when pre-trained on image and text pairs with contrastive learning objectives. We hypothesize that the visual concepts, if captured by the pre-trained VLMs, can be directly extracted by prompting their vision-language interface, without needing to probe their internal representations. Our research question can thus be formulated as discovering the visual concepts encoded by pre-trained VLMs, and evaluating the quality of the extracted concepts.

Interestingly, several recent works (Menon & Vondrick, 2022; Yang et al., 2023b; Yun et al., 2023) reached different conclusions on the encoding of visual concepts in pre-trained VLMs. For example, Yun et al. (2023) observed that CLIP does not appear to recognize fine-grained visual attributes for birds (Wah et al., 2011), where the list of visual attributes is pre-defined by bird experts. In contrast, Menon & Vondrick (2022) demonstrated that object prompts with visual concepts proposed by a large language model (LLM) appear to provide interpretable object classification, as the concept descriptions are nicely correlated with the recognized object categories. As illustrated in Figure 1, we attribute these discrepancies to their different strategies for extracting visual concepts in a pre-trained VLM: First, according to the first and second columns of Figure 1, certain *shortcuts* (e.g., the object category *California seagull*) may dominate the text prompts when recognizing visual concepts, reinforcing the object-concept correlations as given by a prior knowledge base, such as LLMs. A possible remedy is to discover general visual concepts shared by multiple objects, without including the shortcuts in a text prompt. Second, according to the third column in Figure 1, some of concepts proposed by LLMs are not visually discriminative (e.g., "a nocturnal bird") or cannot be reliably recognized by VLMs (e.g. concept prompts marked in red, or the expert prompts used by Yun et al. (2023)). They should be excluded from the list of visual concepts. More generally, in order to draw a convincing conclusion on whether pre-trained VLMs learn to encode visual concepts, one should discover the list of visually discriminative concepts and use them to prompt VLMs in a shortcut-free manner, and then measure the quality and thoroughness of the discovered concepts, when they are tasked to recognize fine-grained objects and generate interpretable explanations with visual concepts.

In this paper, we propose a novel concept discovery and learning framework that extracts *shortcut*-free and visually discriminative concepts from pre-trained VLMs. To discover general concepts that do not include category specific *shortcuts*, we propose to utilize a large and diverse image captioning dataset to discover

general concepts shared by multiple objects. To select concepts that are well-recognizable by VLMs, we propose to jointly consider LLM and VLM information during concept discovery. We prefer the visual concepts that can be both reliably recognized from the images by a VLM, and deemed as suitable descriptions for the objects of interest in the images by an LLM. Furthermore, although visual concepts can be discovered from frozen pre-trained VLMs, they are not always perfectly aligned with images (see Column 2 in Figure 1). To improve the quality of selected concepts, we propose a self-supervised method to fine-tune the last projection layer of VLMs by leveraging the knowledge already encoded in pre-trained VLMs and LLMs. We name our overall framework as Concept Discovery and Learning (CDL).

We design a suite of evaluation protocols for the quality of the extracted concepts. Intuitively, the desirable visual concepts should be precise, to faithfully reflect the visual concepts presented in an image; They should also be thorough, so that the concepts can provide most of the distinctive visual features for an object of interest. Furthermore, these concepts should be generalizable, to effectively describe new and unseen objects. According to the experimental results, the proposed CDL framework effectively discover and learn precise, thorough and generalizable concepts. The discovered concepts lead to accurate and interpretable object recognition on nine visually diverse benchmarks. The results demonstrate that pre-trained VLMs can indeed learn visual concepts via its vision-language interface, and these visual concepts can lead to performant and interpretable visual recognition.

## 2 Related Work

Vision-and-language models (VLMs) pretrained on unlabeled pairs of images and texts from the internet have shown great success on multi-modal benchmarks. Based on the pre-training objectives, VLMs can be broadly categorized into two primary types: contrastive VLMs and generative VLMs. Contrastive VLMs (Radford et al., 2021; Alayrac et al., 2022; Jia et al., 2021; Yao et al., 2021) focus on learning joint representations of images and texts by leveraging contrastive learning techniques (Chen et al., 2020). Generative VLMs Lu et al. (2019); Wang et al. (2023); Chen et al. (2023); Liu et al. (2024) aim to generate coherent text descriptions from images. Representations learned by these VLMs can be transferred to a wide range of tasks, such as image classification (Pratt et al., 2023), visual question answering (Li et al., 2023; Bai et al., 2023), and image and video captioning (Zhang et al., 2021; Yang et al., 2023a).

Visual concepts, which represent the fundamental factors of variations (e.g. colors, shapes) (Hu et al., 2018) in the visual world, have been widely utilized to develop intepretable visual models that can generalize compositionally (Farhadi et al., 2009; Lampert et al., 2009; Nagarajan & Grauman, 2018; Stein et al., 2024). Previous works (Sun et al., 2015; Hernandez et al., 2021; Lee et al., 2023) have proposed to discover textual terms associated to visual concepts to interpret visual systems. These concepts can be integrated into neuro-symbolic frameworks (Mao et al., 2019; Yi et al., 2018) and offer concept-based explanations for visual recognition (Koh et al., 2020). Contrastive VLMs, learning joint representation for images and texts, provide a natural interface for discovering such visual concepts. However, it remains unclear whether VLMs pretrained with contrastive objectives can learn discoverable visual concepts. Previous studies have explored the capability of VLMs to capture and compose primitive concepts (Yun et al., 2023; Yuksekgonul et al., 2022a) and bind visual concepts with objects (Lewis et al., 2022; Yamada et al., 2022). Yun et al. (2023) demonstrate that VLMs do not capture composable primitive concepts by intervening a linear classifier learned from VLM-predicted concepts. Meanwhile, previous works (Pratt et al., 2023; Menon & Vondrick, 2022) show that concept-augmented prompts do provide improvements for VLM-based image recognition. In this paper, we aim to address this discrepency and understand pretrained VLMs' true capability of encoding interpretable visual concepts.

To interpret the decision of visual models with visual concepts, Koh et al. (2020) proposed Concept Bottleneck Model (CBM) to decompose the visual recognition into concept classification and concept-category mapping. While an alternative approach (Bau et al., 2017; Kim et al., 2018; Hernandez et al., 2021) aims to develop post-hoc interpretation for visual models by directly analyzing the internal representation of the models within the concept space, the CBM does not rely on the model having already learned concepts, instead, it simultaneously trains the concept classifier and linear concept-category mapper upon the model being analyzed with the image classification objective. As a result, the CBM can evaluate the concept learning

ability of a model through the concept-category map, regardless of whether the model was previously trained to learn those concepts. Recent works have developed various methods to enhance the performance and adaptability of CBMs (Yuksekgonul et al., 2022b; Hu et al., 2024) and have applied them to machine learning tasks that require high transparency, such as medical diagnosis (Yan et al., 2023b; Chauhan et al., 2023). However, previous CBM-based approaches typically rely on concept classifiers trained with supervision. In this paper, we propose to build CBMs based on the pre-trained VLMs to evaluate their concept learning ability. Assuming that pre-trained VLMs inherently learn visual concepts, we directly extract concept-image similarities from the VLM encoders to serve as the concept classifier and only train a one-layer concept-to-category mapper.

## 3 Method

We first describe a framework for extracting visual concepts from pre-trained VLMs and its application for object recognition in Section 3.1. We show in Section 3.2 that concepts which are non-visual or include certain *shortcuts* might lead to wrong correlations, where the concept activations do not correspond to how likely the visual concepts are actually present in an image. In Section 3.3, we propose a concept discovery method to select *shortcut*-free visual concepts from a large image captioning dataset by utilizing VLM and LLM knowledge to evaluate the visual discrimination. We also propose a self-supervised learning framework to re-align the image-text interfaces of VLMs to improve the quality of selected concepts. In Section 3.4, we describe the method to select a compact set of concepts for specific object recognition tasks. In Section 3.5 we propose a suite of quantitative and human evaluations on the interpretability, precision, thoroughness, and generalizability of the discovered visual concepts. Together they help us understand if pre-trained VLMs learn to encode visual concepts.

### 3.1 Object Recognition via Visual Concepts

Vision-language models such as CLIP (Radford et al., 2021) jointly learn an image encoder $\mathcal{E}_\mathcal{I}$ and a text encoder $\mathcal{E}_\mathcal{T}$ to align images and texts in a shared embedding space. Thanks to the flexibility of the image-language interface, several recent works (Pratt et al., 2023; Menon & Vondrick, 2022; Yun et al., 2023; Yang et al., 2023b; Yan et al., 2023a) attempted to construct semantically interpretable representations by projecting an encoded visual embedding with basis defined by encoded text embeddings. The text embeddings are obtained by encoding manually designed or automatically generated text "prompts" that are likely to correspond to visual concepts. Concretely, given an image $I$ and a set of concept prompts $\mathbb{P}$ of $N$ concepts, one can project the visual features $\mathcal{E}_\mathcal{I}(I)$ into the space of concepts as an $n$-dimensional concept activation vector $\mathbf{a} = [a_1, a_2, ..., a_N]$. Each concept activation is computed as $a_i = \text{sim}(\mathcal{E}_\mathcal{I}(I), \mathcal{E}_\mathcal{T}(p_i))$ (1), where $p_i$ is the $i$-th concept prompt, and $\text{sim}(\cdot)$ is a similarity function, such as cosine similarity, that measures the similarity between the encoded visual and text embeddings. The activations are standardized with z-score normalization to ensure comparability across objects. Our paper assumes that the visual concepts, if well learned by a VLM, can be extracted as concept activations $\mathbf{a}$ via the text prompts.

The concept activations can be utilized for multi-modal object recognition with a function $f : \mathbb{R}^N \to \mathbb{R}^M$ that predicts object categories, where $M$ is the total number of categories. When $f$ is a linear function $f(\mathbf{a}) = \mathbf{a} \cdot \mathcal{W}$, the learned $N \times M$ weight matrix $\mathcal{W}$ allows us to interpret how the visual concepts are utilized for object recognition. A higher positive weight $w_{ij}$ indicates the $i$-th concept is deemed as important positive evidence for recognizing the $j$-th object category, and a near-zero weight indicates the concept is deemed as irrelevant. The linear classifier that maps concept activations to categorical predictions is referred to as Concept Bottleneck Models (Koh et al., 2020) (CBM), which have been adopted to study concept learning with VLMs in Yang et al. (2023b); Yun et al. (2023); Yan et al. (2023a). For zero-shot object recognition when $f(\cdot)$ cannot be learned from data, some works (Pratt et al., 2023; Menon & Vondrick, 2022) assume the visual concepts are category-specific (hence the object names are included in the prompts), and simplify $f(\cdot)$ to be a linear function $f_j(\mathbf{a}) = \frac{1}{|\mathcal{C}_j|} \sum_{a_i \in \mathcal{C}_j} a_i$, where $\mathcal{C}_j$ is the set of visual concepts for the $j$-th category, and $\mathcal{C}_j \cap \mathcal{C}_k = \emptyset$. We observe that this assumption does not hold for real-world datasets as many visual concepts are often shared by several objects, as illustrated in Figure 1.

### 3.2 Do Prompted Activations Correspond to Visual Concepts?

Large language models (LLMs) are often utilized as the knowledge source to propose the relevant visual concept prompts given an object category (Pratt et al., 2023; Menon & Vondrick, 2022; Yang et al., 2023b; Yan et al., 2023a). One example is illustrated in Figure 1, where for California Gull, concept prompts such as "California Gull, which has a long, black tail" are proposed. As discussed in Section 3.1, the linear function $f(\mathbf{a})$ allows us to identify the most important visual concepts for object recognition by picking the highest weighted $w_{ij}$ concepts $i$'s for object $j$. One can then consider two proxy evaluations to measure whether the prompted concept activation $a_i$ actually corresponds to the visual concept $i$ that is present in an image: First, by measuring the object classification accuracy, and assuming that the higher the accuracy is, the more precise the concept activations are. Second, by comparing the top ranked concepts for an image of a certain object category and those identified by human experts, which allows us to understand not only the precision but also the thoroughness of the concepts.

We observe that these proxy evaluations, while intuitive, require careful design of concept prompting strategy in order to draw conclusions on whether the concepts are actually encoded by the pre-trained VLMs. The first issue we identify is the existence of certain shortcuts in the text prompts, leading to "false positive" conclusions. For example, Figure 1 illustrates the concepts with the highest activations according to CLIP. Although the first column appears to indicate that most of the selected concepts are semantically correlated with the input image of a California Gull, it remains unclear whether the concepts are retrieved because they are recognized by CLIP or the class name is utilized as a shortcut. We perform a simple ablation to investigate this potential shortcut: In the second column, we observe that when we combine the category names with randomly shuffled descriptors, CLIP tends to align images with concepts that contain correct category names and wrong descriptors. This phenomenon indicates that the category names, instead of the visual concepts, are used to generate the concept activations. We validate these qualitative observations in Section 4.2, where we demonstrate consistently across nine datasets that the zero-shot classification accuracy remains similar when LLM-generated concepts or randomly shuffled concepts are paired with category names, and that the accuracy drops significantly when category names are removed from the text prompts.

A second issue is the existence of sub-optimal text prompts, leading to "false-negative" conclusions. For example, the concepts marked with purple background in Figure 1 are not visually recognizable (e.g. "nocturnal bird"), and hence should not be considered as visual concepts. Similarly, we observe that the text prompts used by Yun et al. (2023) are designed by birding experts, which may not be in a friendly format to serve as text prompts (e.g. "crested head pattern").

In order to address both issues, we propose to discover category-generic (hence no shortcuts) and visually discriminative concepts directly from a pre-trained VLM (hence the text prompts are more friendly to the VLM). We then propose a suite of evaluations in order to draw robust conclusions on the precision and thoroughness of the discovered concepts.

### 3.3 Visual Concept Discovery and Learning

We propose to discover visual concepts from a large pool of diverse objects, so that the discovered concepts would be shared by multiple objects and generalizable to different domains. Towards this goal, we adopt the Conceptual Captions 3M (Sharma et al., 2018) dataset (CC-3M). CC-3M contains three million images and their captions, hence offering a huge repertoire of objects and their characteristics rendered in both images and text. As illustrated in Figure 2, our overall goal is to identify the key objects and their associated visual concepts from the image captions, with the help of a large language model as the knowledge source. The candidate visual concepts are then checked for their visual discriminativeness, by verifying if a candidate visual concept can be reliably recognized from the corresponding image. Thus, we can select the concepts that can both be recognized by the VLM in the image and be deemed by the LLM as a relevant attribute for the object described in the caption.

As objects in captions often hold specific dependencies, such as serving as the subject or object of an action, we employ Dependency Parsing (De Marneffe et al., 2006) along with a set of designed rules (see Appendix for details) to retrieve the words and phrases that potentially correspond to objects in the captions (e.g.,

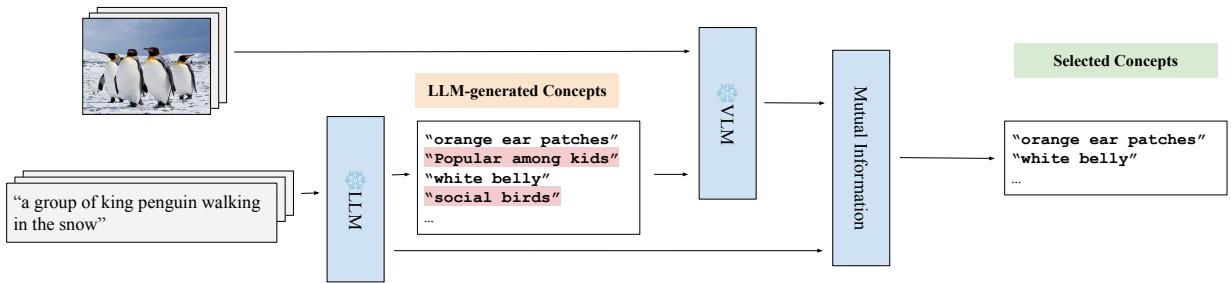

Figure 2: Illustration of our proposed concept discovery method. Given image-caption pairs, we first identify objects from the captions and utilize a large language model to propose candidate concepts for the objects. The concepts are then ranked by the agreement between VLM knowledge (concept recognition from the image) and LLM knowledge (concept proposed based on the caption) based on mutual information.

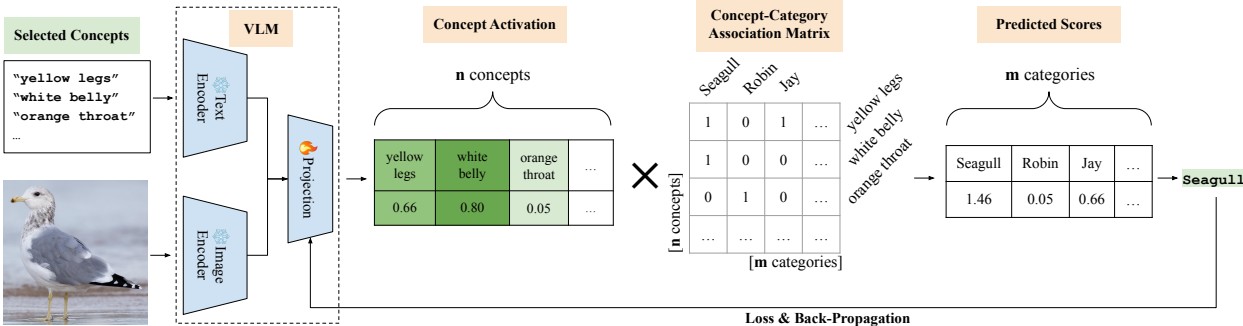

Figure 3: Illustration of the concept-based object recognition framework and our proposed concept learning method. We map the concept activations $\mathbf{a}$ to categories with the concept-category association matrix $\mathcal{W}$. For object recognition, only $\mathcal{W}$ is optimized based on object classification supervision. For concept learning, we assume $\mathcal{W}$ is a binary matrix given by LLM knowledge, and learns to update $\mathbf{a}$ by fine-tuning the last layers of visual and text encoders in the VLM. We rely on the VLM recognized object labels as opposed to ground truth object labels, hence the process is self-supervised.

"king penguin" from a sentence "a group of king penguins walking in the snow"). We then leverage the LLM as an external knowledge base to obtain visual concepts for recognizing the objects. We design a set of prompts (e.g., "What are useful visual features for distinguishing an {object} in a photo?") to query the LLM and retrieve relevant concepts for the object of interest. We take the union of the concepts discovered for all objects as the preliminary list of generic visual concepts.

We then propose to filter the concepts so that the selected ones are also visually discriminative. Intuitively, a visual concept should be ranked higher when the VLM can recognize it from the image if and only if when the same concept is proposed by the LLM based on the image caption. Otherwise, the concept either is not specific to the object of interest (i.e. can be recognized from the image but not proposed to describe any object mentioned in the caption), or likely to correspond to non-visual concepts (e.g. "popular among kids" and "social birds" in Figure 2). Specifically, given a concept and a list of image-caption pairs, we define two variables $X$ and $Y$, where $X$ corresponds to the image-concept similarity as measured by a VLM, and the $Y$ is a binary indicator on the caption-concept correspondence according to an LLM. A high $X$ value $x$ indicates that the VLM can recognize the concept from the image well, and a higher $Y$ value $y$ indicates the concept is deemed by the LLM as a relevant attribute for the object described in the caption. In practice, we compute $x$ as the cosine similarity between the CLIP embedding of the concept and the image, and define $y$ as a boolean indicating whether the concept is relevant to the object in the image, as determined by the

LLM. We adopt the Mutual Information (MI) to determine the agreement between the VLM and the LLM:

$$I(c) = \sum_{y \in Y_c} \sum_{x \in X_c} P_{x,y}(x,y) \log \frac{P_{x,y}(x,y)}{P_x(x)P_y(y)}, \tag{2}$$

which measures the amount of information gain of one variable by knowing another variable. A higher $I(c)$ means that $X_c$ and $Y_c$ are in agreement.

Although we can discover general and visually discriminative concepts from pre-trained VLMs, these concepts are not always perfectly-align with images as shown in Column 2 of Figure 1 and Row 4 of Table 1. We propose a self-supervised fine-tuning method to refine the image-concept alignment of VLMs by leveraging the knowledge already encoded in pre-trained VLMs and LLMs. We first perform the zero-shot classification with pre-trained VLMs to generate pseudo labels for training. We then construct a "ground-truth" concept-category association matrix $\mathcal{W}_{\mathrm{LLM}}$ with binary weights, where each weight is obtained by querying the LLM with prompts like "Does the {category_name} usually have the attribute {concept}?". As illustrated in Figure 3, we project the image and concept embeddings into concept activations $\mathbf{a}$ using the linear projection. The prediction of image category is obtained by the matrix multiplication, where $f(\mathbf{a}) = \mathbf{a} \cdot \mathcal{W}_{\mathrm{LLM}}$. Assuming that the $\mathcal{W}_{\mathrm{LLM}}$ provides ground-truth concept-category association, the loss derived from the predicted labels is utilized to optimize the image-concept alignment. The whole framework is self-supervised as it requires no additional human annotations.

### 3.4 Visual Concept Applications

Our concept discovery and learning framework aims to obtain a list of generic and visually discriminative concepts. We propose to apply the discovered concepts on fine-grained object recognition benchmarks such as bird classification (Wah et al., 2011) and food classification (Bossard et al., 2014) to evaluate their quality. For such applications, it is often desirable to obtain a compact and performant set of visual concepts (e.g. "four-wheeled" is useful for recognizing cars, but not so much for birds). Towards this goal, we first construct the concept-category association matrix $\mathcal{W}_{\mathrm{LLM}}$ and perform concept learning only for the object categories in the target dataset, so that irrelevant concepts not used by any objects are automatically discarded.

The list of remaining visual concepts can be further optimized based on their usefulness and generalizability. We first try to identify the visual concepts that can be reliably recognized from the target dataset. We re-purpose $I(c)$ where $Y_c$ is now obtained by looking up the concept association matrix $\mathcal{W}_{\mathrm{LLM}}$ knowing the ground truth object label for an image. A higher $I(c)$ means the concept $c$ is useful to recognize a subset of object categories in the dataset (according to the LLM knowledge used to construct $\mathcal{W}_{\mathrm{LLM}}$), and can be reliably recognized from the images when the concept is expected to appear.

We also expect the selected concepts to be generalizable, that is, to benefit the recognition of unseen objects. We employ a heuristics where a visual concept is more likely to generalize if it is already used by many known object categories. The "generalizability" of a concept $G(c)$ can hence be estimated by the ratio of object categories that contain the concept $c$ over the total number of object categories.

There exists a natural trade-off between the usefulness and generalizability of a visual concept, we hence compute the weighted average $\alpha \cdot I(c) + (1 - \alpha) \cdot G(c)$ to rank the concepts and apply a fixed budget on the number of visual concepts to use for each downstream benchmark. We select $\alpha$ based on classification performance on the validation set (a lower $\alpha$, namely more general concepts, is preferred whenever the accuracy remains high).

### 3.5 Visual Concept Evaluation

Finally, we propose a suite of evaluation protocols to measure the quality of VLM-discovered concepts in terms of *Interpretability*, *Precision*, *Thoroughness* and *Generalizability*.

**Interpretability** metric from Yun et al. (2023) quantitatively measures how well a VLM learns concept-category associations. The discovered concepts are interpretable if they are associated with images in a human-understandable manner. For example, we expect the model to recognize an image of a "Giant

Panda" according to the concepts "Rounded head", "Black eye patches" and "Black and white fur" instead of irrelevant concepts like "long wings". In this way, the discovered concepts should lead to a concept-category association matrix that is similar to $\mathcal{W}_{LLM}$. Given a trained concept-category matrix $\mathcal{W}_{trained}$, we can predict the category for an image with its concept activations, that is, $f(\mathbf{a}) = \mathbf{a} \cdot \mathcal{W}_{trained}$. Following Koh et al. (2020); Yun et al. (2023), we intervene the classification with ground-truth concept activations $\mathbf{g}$, where $\mathbf{g}_i = 1$ if the ground-truth category of the image contains the $i$-th concept according to the LLM knowledge and $\mathbf{g}_i = 0$ otherwise. We calculate the accuracy of predictions made by $f(\mathbf{g}) = \mathbf{g} \cdot \mathcal{W}_{trained}$ as the intervention accuracy. High intervention accuracy indicates high interpretability of visual recognition since the concept-category association is consistent with human knowledge.

**Precision** is measured with human evaluation on how well the discovered concepts provide correct reasoning clues for visual recognition. As illustrated in Section 3.1, the contribution of a certain concept to the predicted category $j$ of an image can be measured by the dot product between the concept activation $\mathbf{a}$ and $\mathcal{W}_j$ from a CBM weight matrix $\mathcal{W}$. We expect the top-weighted concepts to be accurate depictions of the object of interest in the image when the model prediction is correct. For each correctly-classified image, we ask human annotators to determine whether the top-$k$ weighted concepts actually describe the image and calculate the proportion as the precision score. The selection of the parameter $k$ is contingent upon the scale of the concept space within each dataset. For datasets with large amounts of concepts, we can find more descriptive concepts for a certain category. Hence, we set a larger value for $k$. Conversely, datasets with relatively few concepts have a diminished pool of descriptive elements for individual categories. Consequently, a smaller value for $k$ is deemed appropriate in such scenarios.

**Thoroughness** measures how well the discovered concepts cover important features to recognize the objects in downstream domains. Given an image correctly predicted for its category, we first leverage the LLM knowledge to obtain a complete list of concepts that are potentially related to the category. We then ask human annotators to select from the complete list of concepts that can be inferred visually from the image of interest. We call these the important visual concepts for an image. We then select the top-$k$ weighted concepts and calculate the thoroughness score as the percentage of important visual concepts covered by the top-$k$ weighted concepts.

**Generalizability** measures whether the discovered and learned concepts can benefit the recognition of unseen objects. We consider in-domain and cross-domain generalization of the discovered concepts. For the in-domain generalization, we first randomly split the category list of a given dataset into seen categories and unseen categories. We select from the discovered concepts separately for the seen and unseen categories (as in Section 3.4), we then perform concept learning only the examples of seen categories. These concepts are transferred to train concept-based classifiers for the unseen categories, and we report classification accuracy as the proxy to measure in-domain generalization. Cross-domain generalization is measured following the same procedure, but for a seen dataset and a different dataset.

# 4 Experiments

We conduct quantitative and human evaluations to evaluate the discovered and learned concepts. In Section 4.2, we demonstrate why we need short-cut free and visually discriminative concept discovery and learning by showing that concept prompts in previous works might be category-biased. In Section 4.3 and Section 4.4, we aim to prove that pre-trained VLMs do learn visual concepts by demonstrating that the discovered concepts can lead to accurate classification and exhibit high qualities including interpretability, precision, thoroughness and generalizability. In Section 4.5 we aim to explore how specific components in our proposed framework contribute to the quality of discovered concepts.

## 4.1 Experimental Setup

**Datasets:** We conduct experiments on several challenging fine-grained image classification datasets, including ImageNet (Deng et al., 2009), Food (Bossard et al., 2014), CIFAR-100 (Krizhevsky et al., 2009), CIFAR-10, CUB (Wah et al., 2011) ,Flowers (Nilsback & Zisserman, 2008) Stanford Cars (Krause et al.,

| Prompt Design | ImageNet | Food | CIFAR-100 | CIFAR-10 | CUB | Flowers | Stanford Cars | Aircrafts | Oxford Pets |
|---|---|---|---|---|---|---|---|---|---|
| Category Name | 71.6 | 91.8 | 75.9 | 96.2 | 63.1 | 77.4 | 77.3 | 36.1 | 93.5 |
| Name w/ LLM Concepts (Menon & Vondrick, 2022) | 75.0 | 92.4 | 77.7 | 96.6 | 63.5 | 78.9 | 77.6 | 37.4 | 92.2 |
| Name w/ Random Concepts | 70.1 | 91.6 | 75.4 | 95.0 | 62.1 | 78.7 | 76.5 | 36.0 | 92.4 |
| LLM Concepts only | 22.1 | 3.6 | 30.9 | 70.7 | 5.3 | 7.0 | 3.2 | 1.6 | 6.4 |

Table 1: Zero-shot classification on nine object recognition benchmarks. We observe that although augmenting category names with LLM-generated concepts improves classification accuracy (e.g. VDES (Menon & Vondrick, 2022)), the method still mainly relies on category names in the text prompts, and it remains unclear if the concepts included in the text prompts are properly recognized by pre-trained VLMs. When paired with randomly shuffled concept in the prompts, the accuracy drops only moderately; when the category names are removed, the accuracy drops significantly.

2015), Aircrafts (Maji et al., 2013) and Oxford Pets (Parkhi et al., 2012). The statistics and split of the datasets are shown in the appendix.

**Baselines:** We compare with LaBo (Yang et al., 2023b) and LM4CV (Yan et al., 2023a), which are the state-of-the-art works on concept-based visual recognition. Following the setting in LM4CV, we control the bottleneck sizes (number of concepts) to be the same for the baselines and our model for fair comparison. Besides full-shot classification, we also compare with LaBo on few-shot settings, where we select concepts and train the model on limited data. It is infeasible for LM4CV to conduct few-shot classification since it requires the whole training set on concept generation.

**Implementation Details:** We use the same LLM `GPT-3-text-davinci-002` to obtain descriptors as previous works. We also use the same CLIP backbone (ViT-L-14) to compare with baseline models. Following (Yun et al., 2023), we use logistic regression to train the concept bottleneck models. We observe that the performance of CBM is robust to the choice of hyperparameters and use the default Scikit-learn hyperparameters for all datasets. For concept learning, we use the AdamW optimizer with 5e-4 learning rate and 1e-4 weight decay to fine-tune the CLIP model, and we use the validation loss to select checkpoints. For human evaluation experiments, we hire annotators from Amazon Mechanical Turk. For each example, we ask three annotators to annotate and use the majority vote to obtain the result. We conduct Students' T-test (Student, 1908) to evaluate the statistical significance of the human evaluation results. The detailed design of human evaluation and the statistical significance results are shown in the appendix.

### 4.2 Concept-Augmented Prompts Are Category-Biased

As discussed in Section 3.2, we conduct ablation experiments to understand if the concepts used in the prompts of previous zero-shot classification methods lead to improved classification accuracy. The results in Table 1 are consistent with Figure 1, both of which show that the concept-augmented text prompts do not offer conclusive evidence whether pre-trained VLMs learn to encode concepts. Even random concepts have minimal impact on zero-shot performance (as shown in Row 2 and Row 3). In contrast, the removal of category names results in a catastrophic decline in zero-shot performance (as shown in Row 2 and Row 4). These observations suggest that category names act as critical shortcuts in concept-augmented prompts.

Besides category biases, the discovered concepts in previous works contain many non-visual concepts as illustrated in Figure 1. We conduct human evaluation to compare our discovered concepts with previous works by the proportion of non-visual concepts and concepts containing class names. For both baselines and our CDL, we randomly select 100 concepts for each dataset for human evaluation. The results are shown in Table 2. We can observe that CDL offers visual concepts that are more category-agnostic and visually discriminative.

### 4.3 Classification Performance with Visual Concepts

Following the settings in LaBo and LM4CV, we set the bottleneck size to be the same or 2 times of number of classes in a given dataset. The same ViT-L-14 CLIP model is used across all methods. To ensure a

|  | %Category-agnostic | %Visual |
|---|---|---|
| LaBo | 65.50 | 66.83 |
| ML4CV | 82.33 | 73.83 |
| CDL | **92.33** | **85.50** |

Table 2: Human evaluation results on the proportion of category-agnostic and visually-discriminative concepts. The results are averaged over all datasets. The results show that our concepts discovered from general corpora are less biased by category names and more visually discriminative.

|  | ImageNet | | Food | | CIFAR-100 | | CIFAR-10 | | CUB | | Flowers | | Stanford Cars | | Aircrafts | | Oxford Pets | |
|---|---|---|---|---|---|---|---|---|---|---|---|---|---|---|---|---|---|---|
| #Concepts | 1000 | 2000 | 101 | 202 | 100 | 200 | 10 | 20 | 200 | 400 | 102 | 204 | 196 | 392 | 102 | 204 | 37 | 74 |
| LaBo | 83.2 | 83.6 | 89.8 | 91.1 | 80.5 | 84.1 | 77.8 | 92.2 | 79.7 | 81.3 | 95.5 | 94.9 | 84.7 | 85.6 | 59.3 | 60.7 | 88.3 | 90.7 |
| LM4CV | 83.4 | 83.6 | 90.0 | 91.1 | 81.7 | 84.2 | 80.4 | 92.3 | 77.5 | 80.4 | 94.5 | 93.8 | 84.9 | 85.7 | 59.9 | 61.0 | 88.5 | 90.4 |
| CDL | **83.6** | **84.0** | **94.2** | **94.9** | **83.8** | **85.8** | **82.1** | **93.5** | **83.2** | **83.4** | **96.3** | **95.7** | **86.1** | **86.7** | **60.8** | **62.4** | **90.6** | **92.3** |

Table 3: Comparison with baselines on classification accuracy with different bottleneck sizes. All methods are based on ViT-L-14 CLIP checkpoint.

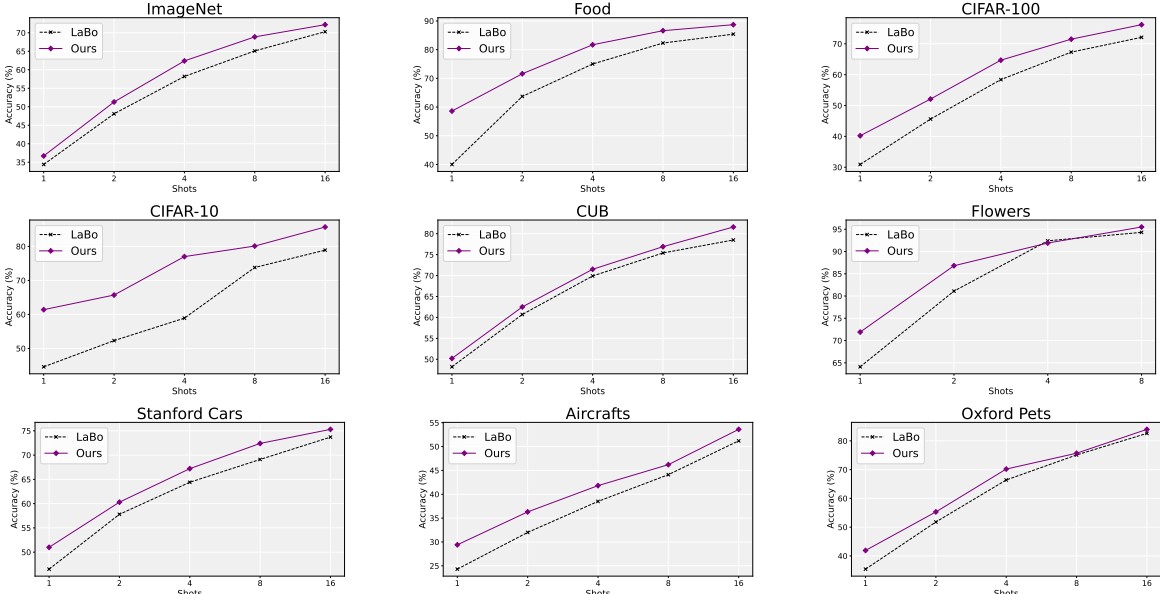

Figure 4: Few-shot classification evaluation with LaBo and our method.

fair comparison, we conduct the classification using our concept discovery and selection methods, without concept learning. Table 3 shows that our method consistently outperforms the baseline methods on all datasets.

We then adopt the few-shot learning setting, where we select the concepts (as described in 3.4) and train the model with the few training examples. Figure 4 shows that CDL consistently outperforms LaBo, especially when the number of training examples is smaller.

## 4.4 Evaluation of the Discovered Concepts

Figure 5 shows the evaluation results of the *Interpretability*, *Precision*, and *Thoroughness* of the baselines and our method. For human evaluation of *Precision*, and *Thoroughness*, we randomly select 100 images from each dataset and ask 3 human workers to annotate the *Precision* and *Thoroughness* of top-weighted concepts for each image as described in 3.5. More details are shown in Appendix Sec. (D). The *Precision*

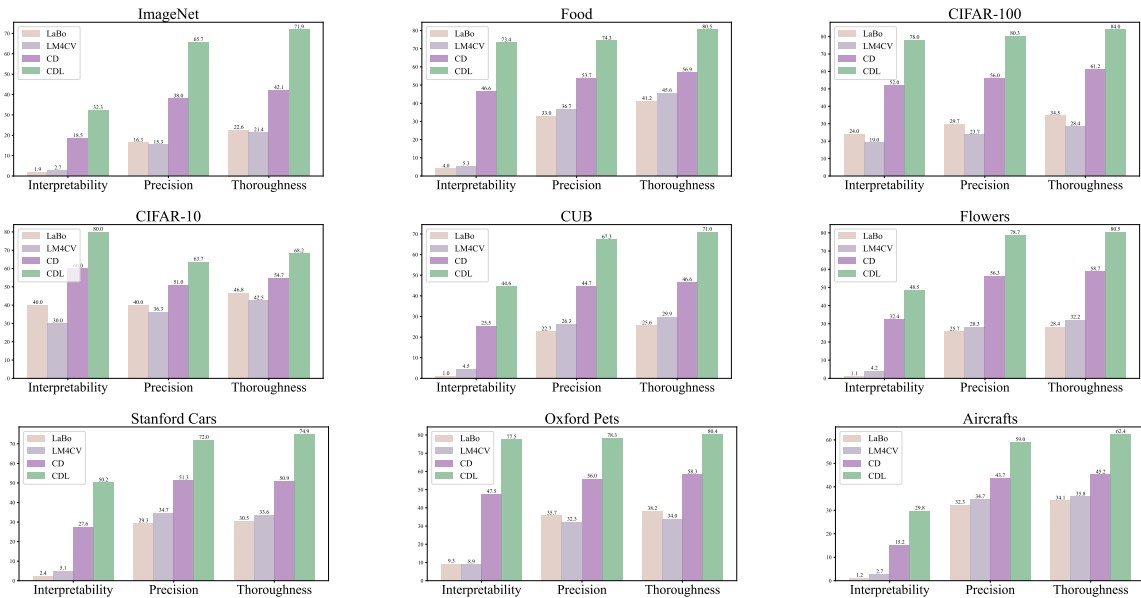

Figure 5: Human evaluation of the discovered concepts for their interpretability, precision, and thoroughness. We evaluate the concepts selected without and with concept learning (denoted as "CD" and "CDL").

| Domain | Method | ΔClassification | ΔInterpretability | ΔPrecision | ΔThoroughness |
|---|---|---|---|---|---|
| In-domain | LaBo | 0.2 | 16.1 | 5.8 | 7.6 |
| | LM4CV | 0.4 | 14.2 | 4.5 | 6.4 |
| | CDL | **1.0** | **18.3** | **7.9** | **10.4** |
| Cross-domain | LaBo | -1.5 | 0.1 | -3.7 | -4.5 |
| | LM4CV | -0.3 | -0.6 | -6.4 | -8.2 |
| | CDL | **0.2** | **7.0** | **10.3** | **13.9** |

Table 4: Generalization evaluation for discovered concepts. The In-domain results refer to improvements on unseen categories in the same dataset, while the Cross-domain results refer to improvements on unseen domains. Δ denotes the improvement of fine-tuned CLIP compared to the original CLIP. Higher improvement means better generalization.

and *Thoroughness* of each dataset are calculated as the averages of these evaluations on the 100 images, with the results displayed on the Y-axis of Figure 5. We can observe that despite having high classification performance, the baseline models discover concepts that exhibit unsatisfactory interpretability, precision, and thoroughness. Our concept discovery method provides significant improvements on all three metrics, and the self-supervised concept learning can further improve the concept quality.

Table 4 shows the in-domain and cross-domain generalization results. Our proposed CDL outperforms both baselines for its generalizability, especially for the cross-domain scenario when transferring from ImageNet to CUB. We observe that LaBo and LM4CV struggle with cross-domain generalization as they select completely different concepts for different datasets and few common patterns can be learned with their methods.

## 4.5 Ablation Study

In this subsection, we perform ablation studies to analyze the performance of different sub-modules in the proposed CDL framework.

**Multi-modal Visual Concept Discovery** To evaluate the effectiveness of the proposed concept discovery method, we build a baseline which only uses the LLM to generate concepts and perform random concept

|  | Accuracy | Interpretability | Precision | Thoroughness |
|---|---|---|---|---|
| LLM-only | 68.5 | 3.6 | 24.2 | 27.0 |
| Multi-modal | **83.4** | **44.6** | **67.3** | **71.0** |

Table 5: Comparison of our proposed multi-modal concept discovery method with the LLM-only concept discovery method on CUB dataset. Our method significantly outperforms LLM-only method on both classification accuracy and concept quality.

| Domain | Source | $\Delta$Classification | $\Delta$Interpretability | $\Delta$Precision | $\Delta$Thoroughness |
|---|---|---|---|---|---|
| In-domain | CUB | 0.2 | 12.5 | 2.8 | 3.7 |
|  | CC-3M | **1.0** | **18.3** | **7.9** | **10.4** |
| Cross-domain | ImageNet | -1.1 | -0.2 | -5.5 | -4.6 |
|  | CC-3M | **0.2** | **7.0** | **10.3** | **13.9** |

Table 6: Comparison of the concept discovered from CC-3M and other datasets on both in-domain and cross-domain generalization evaluation for the CUB task. The concepts discovered from CC-3M have better generalizability in both cases.

selection for downstream tasks. As shown in Table 5, the concepts discovered by the LLM-only method are of unsatisfactory classification accuracy, interpretability, precision and thoroughness, which demonstrates the necessity of utilizing multi-modal information to select visually discriminative concepts. The results show that our proposed multi-modal concept discovery method can effectively improve the classification performance and the interpretability, precision and thoroughness of the discovered concepts.

**Utilization of the CC-3M dataset** To analyze the effectiveness of the utilization of the CC-3M dataset during concept discovery, we perform the proposed concept discovery method on the CC-3M (Sharma et al., 2018) dataset and the downstream datasets. The performances of both methods are similar on CUB (details are shown in Appendix), which indicates that our proposed concept discovery method can still provide significant improvement on the classification and concept quality with only downstream datasets. Considering the cost of human evaluation for the *Precision* and *Thoroughness*, we select the CUB dataset as the downstream benchmark for the generalization evaluation. The in-domain and cross-domain generalization results in Table 6 indicate that the concepts discovered from CC-3M are more generalizable than the concepts discovered from downstream datasets. In conclusion, the proposed concept discovery and learning method provides improvement on classification performance and concept quality regardless of the utilization of datasets, and discovering concepts from the CC-3M datasets can enhance the generalization of concepts.

## 5 Conclusion and Future Work

In this paper, we investigate the question of whether pre-trained Vision-Language Models (VLMs) can encode primitive visual concepts. We first illustrate that category-biased concepts extracted from specific datasets do not offer conclusive evidence of the concept learning capacity of VLMs. To resolve this issue, we design a novel framework to discover category-agnostic and visually discriminative concepts from a large image-caption dataset with the help of VLMs and LLMs. To make use of the discovered concepts for downstream tasks, we propose a novel method to build concept bottlenecks with a compact and performant set of concepts for a specific domain. We also propose a self-supervised concept learning framework to re-align the concept knowledge in VLMs for the category classification in specific domains. To prove that VLMs do learn useful and interpretable concepts, we propose a suite of comprehensive protocols to evaluate the quality of the discovered concepts and perform exhaustive experiments including human evaluation. The experimental results demonstrate that VLMs do capture primitive concepts that can lead to effective, interpretable, and generalizable visual recognition.

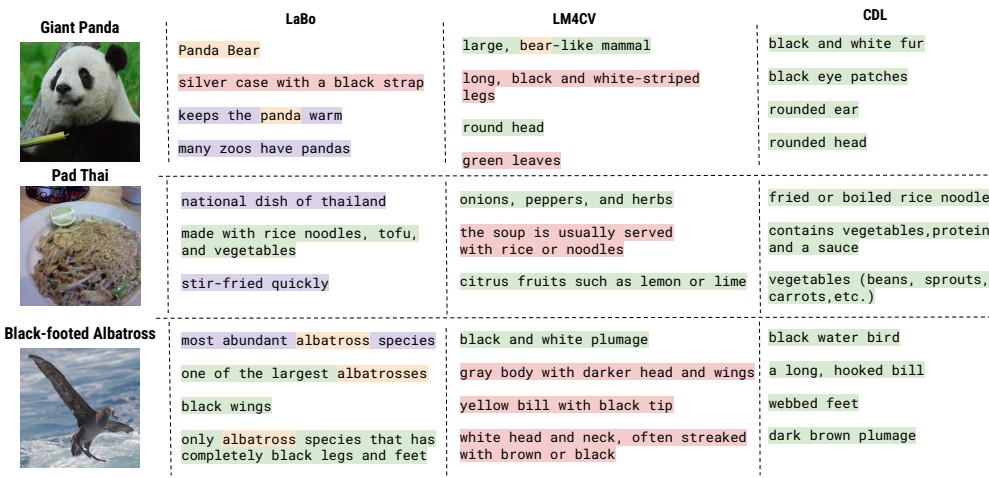

Figure 6: Examples of the top weighted concepts correlated with the given images.

While we illustrate that VLMs do learn discoverable concepts, it is still significant to understand what kind of concept and compositionality knowledge cannot be learned in the contrastive learning-based pre-training of VLMs. In future work, we plan to explore whether VLMs can capture the semantic and spatial relationships between concepts and utilize these relationships to perform complex multi-modal reasoning.

## 6 Acknowledgement

This work is in part supported by a gift from Adobe Research, a seed grant from NASA, and a Richard B. Salomon award for Chen Sun. We thank helpful discussions with Professors Ellie Pavlick and Stephen Bach. We thank Yue Yang, An Yang, Yu Wang for providing the open-sourced code for the baselines.

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

## A  Hyperparameters

We adjust hyperparameters according to the performance of the models on the validation dataset (For ImageNet, we randomly select 10% of the training set as the validation set). For $\alpha$ in Equation 2, we set it to 0.7 for ImageNet dataset, 0.8 for Food-101, CIFAR-100, CUB-200 and Flowers-102 datasets and 0.9 for CIFAR-10 dataset. According to Equation 2 in Section 3.4, a smaller $\alpha$ will increase the generalizability but decrease the discriminativeness of the selected concepts, and thus decrease classification performance. To achieve a trade-off between generalizability of the selected concepts and classification performance, we monitor the classification performance when gradually decreasing $\alpha$. We pick the $\alpha$ right before classification performance drops significantly.

## B  Additional Implementation Details

### B.1  Dependency Parsing Rules

Given a caption, we perform Dependency Parsing to extract the grammatical relations from the caption. We extract the nouns or phrases from the following dependencies as potential objects in an image caption:

- *nsubj / nsubjpass*: The *nsubj* (nominal subject) is the noun that performs the main action. For example, in the sentence "the horse is eating grass", "horse" is the nominal subject. The *nsubjpass* (passive nominal subject) is the noun that performs the main action in a sentence of passive voice. For example, in the sentence "The dog is led by a leash", "dog" is the passive nominal subject. We extract the nominal and passive nominal subject as a word potentially corresponding to an object in the image.

- *dobj / iobj*: The *dobj* (direct object) is the noun or noun phrase that receives the action of the verb. For example, in the sentence "the horse is eating grass", "grass" is the direct object. The *iobj* (indirect object) is the noun or noun phrase that indicates to or for whom the action of the verb is done. For example, in the sentence "The man gives the girl a flower", "girl" is the indirect object. We extract the direct and indirect object as a word potentially corresponding to an object in the image.

- *amod*: An *amod* (adjectival modifier) is an adjective that describes a noun (e.g. "black dog", "white bird"). We extract the amod and its object as a phrase potentially corresponding to an object in the image.

- *compound*: The *compound* label indicates the word is part of a compound phrase like "king penguin". Once select a word following above rules, we check whether it is part of a compound phrase. If so, we extract the whole phrase as the object.

### B.2  Mutual Information Implementation

The Mutual Information (MI) measures the information gain of one variable by knowing another variable. Given two variable $X$ and $Y$, it measures the KL divergence between the joint distribution $P(X, Y)$ and the product of the marginal distribution $P(X)P(Y)$:

$$MI = \sum_{y \in Y} \sum_{x \in X} P_{x,y}(x, y) \log \frac{P_{x,y}(x, y)}{P_x(x)P_y(y)}. \tag{3}$$

The MI is high when $X$ and $Y$ are positively or negatively related (e.g. high $x$ value indicates high $y$ value and low $y$ value indicates low $y$ value or vice versa). Given a concept and an image-caption dataset, the $X$ variable corresponds to the image-concept similarity as measured by a VLM, and the $Y$ variable is a binary indicator on the caption-concept correspondence according to an LLM. Namely, a higher $x$ indicates that VLM can recognize the concept from the image well, and a higher $y$ indicates the concept is deemed by LLM as a relevant "attribute" for the object described in the caption. As such, we prefer concepts with higher MI

since it indicates that if a concept can be recognized from the image, the same concept should be relevant for the paired caption, and vice versa. This helps filter out concepts that are non-visual (relevant according to LLM, but cannot be recognized visually by VLM, such as "fish-eating" and "endangered animal"), irrelevant to the objects of interest (can be recognized by VLM, but irrelevant according to LLM, such as concept "yellow-leg" for recognizing food). Figure A1 shows some examples for the MI calculation.

Since the image-concept similarity is continuous, we utilize K-nearest-neighbors-based entropy estimation toolkit in Scikit-learn to perform discretization and calculate the MI.

## C  Dataset Details

The table A1 shows the statistical details of the datasets we choose.

| Dataset | #Class | #Train | #Valid | #Test |
|---|---|---|---|---|
| ImageNet | 1000 | 128,1167 | 50,000 | - |
| Food-101 | 101 | 60,600 | 15,150 | 25,250 |
| CIFAR-100 | 100 | 40,000 | 10,000 | 10,000 |
| CIFAR-10 | 10 | 40,000 | 10,000 | 10,000 |
| CUB-200 | 200 | 4,794 | 1,200 | 5,794 |
| Flowers-102 | 102 | 4,093 | 1,633 | 2,463 |
| Stanford Cars | 196 | 6,515 | 1,628 | 8041 |
| Aircrafts | 102 | 3,400 | 3,400 | 3,400 |
| Oxford Pets | 37 | 1,846 | 1,834 | 3669 |

Table A1: Statistical details of datasets. "#Class" means the number of classifications. "#Train", "#Valid", and "#Test" denote the instance numbers of each dataset respectively. For ImageNet, we randomly select 10% of the training set as the validation set and regard the validation set as the test set.

## D  Human Evaluation Details

We hire workers on https://www.mturk.com to conduct human evaluation. To make our human evaluation more robust, for one data point we ask three human workers to annotate. For the precision and thoroughness metric, we randomly select 100 images on each dataset to evaluate. For CIFAR-10, CIFAR-100, Food-101, Flowers-102 we choose the top 3 concepts to annotate. For ImageNet and CUB-200 we choose the top 5 concepts to annotate. We annotate 2,200 data points for precision and around 3,000 data points for thoroughness. For the visual discriminability and category name containing, we utilize different methods to select 100 concepts for each dataset to evaluate and report the average score. Hence we annotate 1,800 data points for those two task. In the annotation, we randomly shuffle the order of instances to remove possible biases.

In order to validate the effectiveness of our human evaluation, we calculate the pairwise annotator agreement score following LaBo. The average pairwise annotator agreement proportion on all datasets is 69.4%, which is comparable to the 69.8% proportion in LaBo.

We show some examples of our human evaluation interface in Figure A2. The examples shown are for measuring precision of the concepts, and a similar interface is used to measure thoroughness, where the humans are asked to build a complete concept list for an image.

## E  Statistical Significance

We conduct Students' T-test to evaluate the statistical significance of our main experiments including classification accuracy and human evaluation of concept quality. We set the threshold of p-value to be 0.05 following

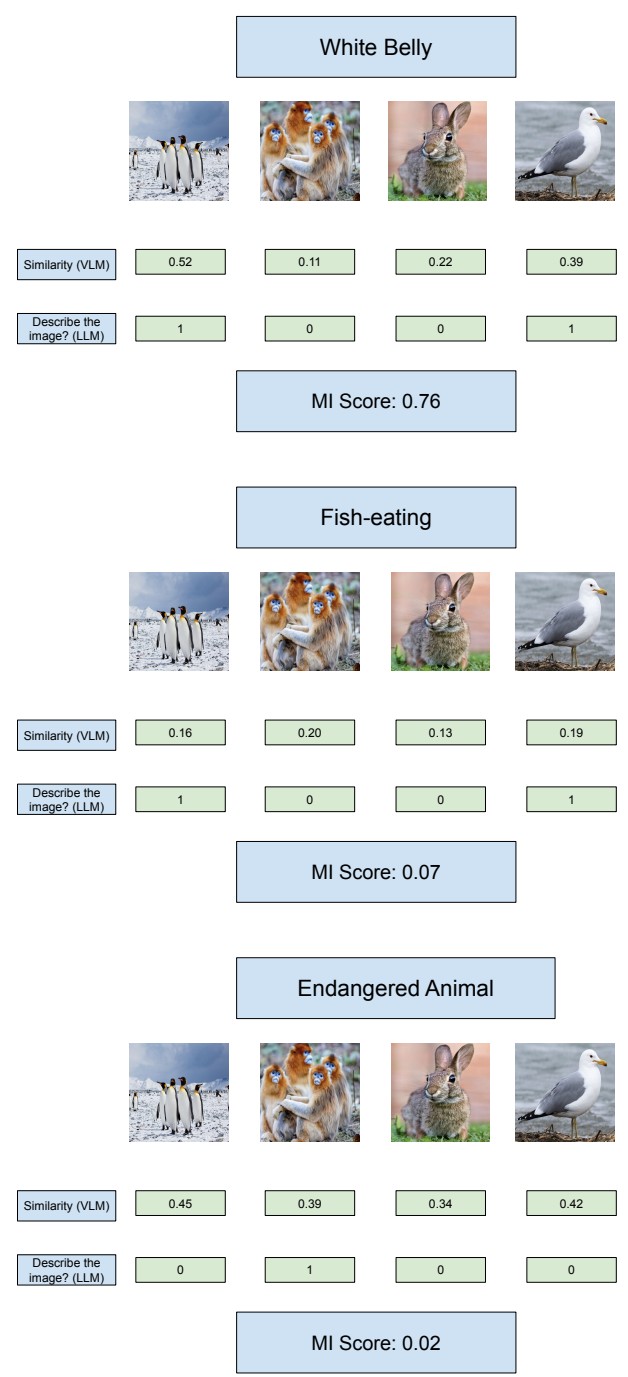

Figure A1: Illustrations of Mutual Information calculation. We can observe that visually discriminative concepts such as "white belly" have high MI score because they can be reliably recognized from images that are supposed to contain the concept according to LLM, and vice versa. The non-visual concepts like "fish-eating" and "endangered animal" have low MI scores because VLMs cannot recognize these concepts from the images that are supposed to contain these concepts according to LLM.

previous works. When p-value is lower than 0.05, the null hypothesis is rejected and out method performs significantly better than the baseline method. From the results in Table A2 we can observe that both our

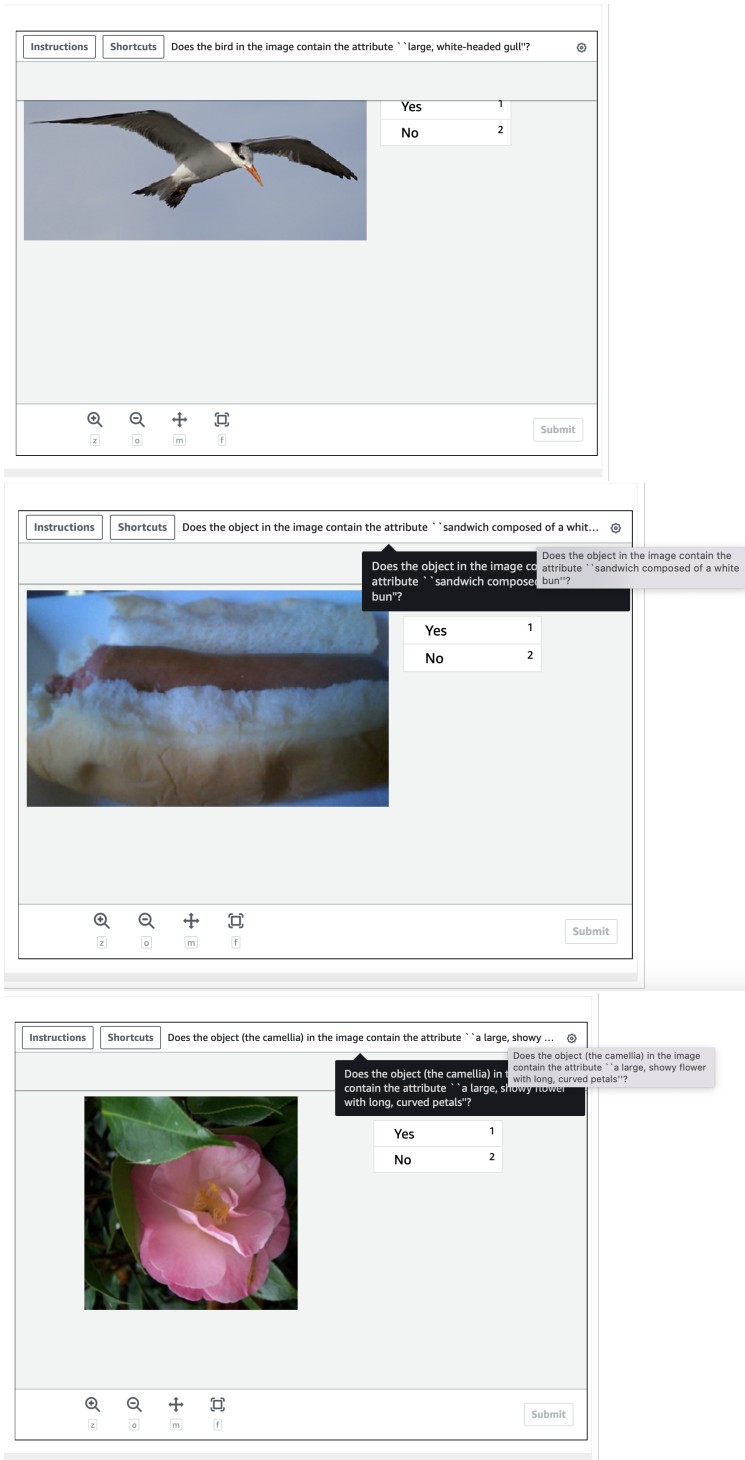

Figure A2: Examples of the annotation interface.

proposed method significantly outperform the baseline methods regarding the classification performance and concept quality.

| Dataset | Method | Precision | | Thoroughness | | Category Agnostic | | Visual | |
|---|---|---|---|---|---|---|---|---|---|
| | | p-value | significance | p-value | significance | p-value | significance | p-value | significance |
| ImageNet | CDL v.s. LaBo | 4.59e-65 | ✓ | 2.67e-62 | ✓ | 3.94e-11 | ✓ | 2.00e-3 | ✓ |
| | CDL v.s. LM4CV | 2.19e-68 | ✓ | 7.02e-66 | ✓ | 3.87e-01 | ✗ | 4.54e-2 | ✓ |
| Food-101 | CDL v.s. LaBo | 7.38e-26 | ✓ | 8.73e-25 | ✓ | 1.32e-05 | ✓ | 1.30e-4 | ✓ |
| | CDL v.s. LM4CV | 1.59e-21 | ✓ | 4.97e-20 | ✓ | 9.31e-05 | ✓ | 2.78e-2 | ✓ |
| CIFAR-100 | CDL v.s. LaBo | 2.85e-40 | ✓ | 3.44e-40 | ✓ | 8.59e-10 | ✓ | 7.26e-3 | ✓ |
| | CDL v.s. LM4CV | 1.27e-51 | ✓ | 4.99e-51 | ✓ | 1.56e-02 | ✓ | 1.78e-2 | ✓ |
| CIFAR-10 | CDL v.s. LaBo | 4.28e-9 | ✓ | 9.58e-8 | ✓ | 5.35e-06 | ✓ | 2.36e-3 | ✓ |
| | CDL v.s. LM4CV | 5.28e-12 | ✓ | 1.39e-10 | ✓ | 7.90e-01 | ✗ | 2.27e-2 | ✓ |
| CUB-200 | CDL v.s. LaBo | 1.31e-50 | ✓ | 4.54e-52 | ✓ | 3.57e-04 | ✓ | 6.18e-4 | ✓ |
| | CDL v.s. LM4CV | 5.12e-42 | ✓ | 2.95e-42 | ✓ | 5.09e-03 | ✓ | 1.40e-1 | ✗ |
| Flowers-102 | CDL v.s. LaBo | 7.44e-45 | ✓ | 3.10e-42 | ✓ | 1.68e-09 | ✓ | 8.44e-3 | ✓ |
| | CDL v.s. LM4CV | 1.21e-40 | ✓ | 4.58e-37 | ✓ | 3.17e-02 | ✓ | 4.09e-2 | ✗ |

Table A2: The statistical significance of the human evaluation results. Smaller p-value indicates higher significance. The results show that out human evaluation results are statistically significant and our discovered concepts are consistently better than concepts in previous works on all datasets.

| | ImageNet | Food | CIFAR-100 | CIFAR-10 | CUB | Flowers | Stanford Cars | Aircrafts | Oxford Pets |
|---|---|---|---|---|---|---|---|---|---|
| CDL v.s. LaBo | 9e-02 (✗) | 2e-74 (✓) | 1e-09 (✓) | 3e-14 (✓) | 1e-06 (✓) | 3e-03 (✓) | 1e-02 (✓) | 2e-01 (✗) | 1e-02 (✓) |
| CDL v.s. LM4CV | 4e-01 (✗) | 1e-68 (✓) | 9e-05 (✓) | 2e-03 (✓) | 1e-14 (✓) | 2e-01 (✗) | 3e-02 (✓) | 4e-01 (✗) | 3e-02 (✓) |

Table A3: The statistical significance of the classification accuracy. Smaller p-value indicates higher significance. The performance gain is significant when p-value is **lower** than 5e-02.

| | ImageNet | | Food-101 | | CIFAR-100 | | CIFAR-10 | | CUB-200 | | Flowers-102 | |
|---|---|---|---|---|---|---|---|---|---|---|---|---|
| #Concepts | 1000 | 2000 | 101 | 202 | 100 | 200 | 10 | 20 | 200 | 400 | 102 | 204 |
| W/o Concept Learning | 83.6 | 84.0 | 94.2 | 94.9 | 83.8 | 85.8 | 82.1 | 93.5 | 83.2 | 83.4 | 96.3 | 95.7 |
| W Concept Learning | 83.8 | 83.9 | 94.4 | 94.9 | 83.6 | 85.3 | 96.1 | 96.5 | 82.5 | 82.1 | 96.6 | 96.2 |

Table A4: Comparison of classification performance with our discovered concepts before and after concept learning. The results show that it is the concept discovery that mainly contribute to the improvement of classification performance.

| | ImageNet | Food-101 | CIFAR-100 | CIFAR-10 | CUB-200 | Flowers-102 |
|---|---|---|---|---|---|---|
| W/o Concept Learning | 21.9 | 55.7 | 62.8 | 60.0 | 25.5 | 27.6 |
| W Concept Learning | 32.3 | 73.4 | 78.2 | 80.0 | 44.6 | 48.5 |

Table A5: Comparison of interpretability of the discovered concepts before and after concept learning. The results of the intervention accuracy show that both concept discovery and learning parts provide significant improvement for the interpretability of the concepts.

# F   Ablation Study on Effectiveness of CDL Framework

In this section, we conduct ablation studies on the effect of different phases of our CDL framework. We compare the classification performance and interpretability of the concept-based image-recognition before and after concept learning to illustrate the effects of concept discovery and concept learning.

The results in Table A4, A5 and A6 indicate that (1) it is the concept discovery that mainly contribute to the improvement of classification performance and the concept learning would not affect the classification accuracy; (2) both concept discovery and learning parts provide significant improvement for the interpretability of the concepts according to the intervention accuracy resutls.

|  | Interpretability | Precision | Thoroughness |
|---|---|---|---|
| W/o Concept Learning | 21.9 | 44.0 | 49.6 |
| W Concept Learning | 32.3 | 65.7 | 71.9 |

Table A6: Human evaluation of the discovered concepts before and after concept learning on ImageNet.

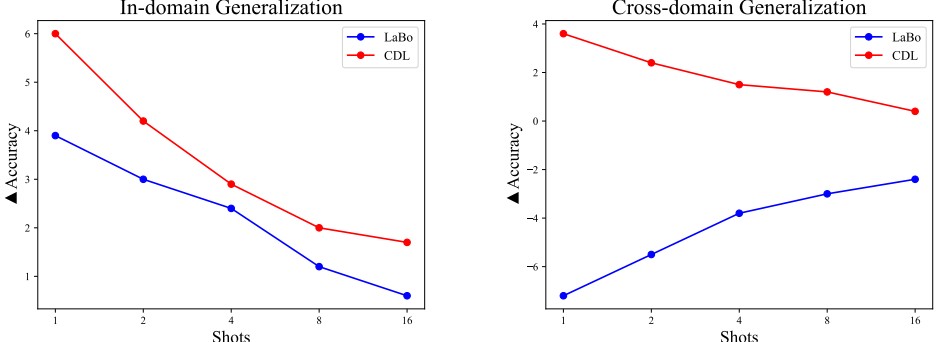

Figure A3: Few-shot classification performance of in-domain and cross-domain generalization, where ▲ denotes the improvement of fine-tuned CLIP compared to the original CLIP. The comparison between our concepts and LaBo concepts suggest that our concepts can provide better generalization and lead to better performance for few-shot recognition on different categories and domains.

| Method | AP | AP50 |
|---|---|---|
| OWT | 43.5 | 64.7 |
| OWT + CDL | **46.2** | **68.1** |

Table A7: Object detection performance on MS-COCO dataset. The baseline results are quoted from Minderer et al. (2022). Visual concept discovery and learning can enhance the object detection performance of the existing framework.

## G  Few-shot Performance about Generalization

To further measure the in-domain and cross-domain generalization of the discovered concepts, we perform few-shot classification with the fine-tuned CLIP model on the unseen categories. The results in Figure A3 show that our discovered concepts can enable much better few-shot classification performance, which suggests that the encoded knowledge of our discovered concepts are more generalizable to unseen categories and domains.

## H  Broader Application of Visual Concepts

Besides image classification tasks, the discovered and learned visual concepts can also benefit tasks that require object localization by localizing visual concepts in images. We integrate the visual concepts into the existing object detection framework OWL proposed by Minderer et al. (2022) and conduct evaluations on the MS-COCO dataset (Lin et al., 2014). We first perform concept discovery and learning on the training data, and then replace the category names (e.g. "dog") with selected visual concepts (e.g. "four-legged mammal" and "floppy ears") to train and evaluate the object detection model. The results are shown in Table A7. The visual concepts can enhance the object detection performance of the existing framework.

