# OpenReview forum: "Pre-trained Vision-Language Models Learn Discoverable Visual Concepts"
_TMLR — Accepted by TMLR_

### Review · Reviewer_cs4X · 2024-10-07

**Summary Of Contributions:**

The work study whether Vision-Language Models (VLMs) inherently learn and encode visual concepts. It proposes a novel Concept Discovery and Learning (CDL) framework to extract generic and visually discriminative concepts, addressing limitations in prior work, i.e., category name bias and lack of visual distinctiveness. The authors argue that previous approaches, while utilising concept prompts, fail to prove whether VLMs actually learn visual concepts or rely on shortcuts, e.g., prompts containing both category names and concepts might achieve high accuracy simply because the VLM recognises the object from its name, not the descriptive concept. The proposed framework tackles these challenges by (i) identifying a diverse set of generic visual concepts from a web-scale image-caption dataset, and (ii) employing a multi-modal approach to select concepts that are relevant to describe an object and reliable to recognise it within an images. Additionally, the authors develop a suite of human evaluation metrics to assess the quality of the discovered concepts in terms of generalisability, precision, interpretability, and thoroughness. Through extensive evaluation on nine benchmark datasets, the authors demonstrate that their approach achieves comparable performance to prior works while yielding concepts that are more aligned with human understanding.

**Audience:**

Yes

**Broader Impact Concerns:**

No concerns

**Claims And Evidence:**

Yes

**Requested Changes:**

- The authors could expand their work by testing other data pools from where to extract the concepts. Currently, it is hard to understand what is the advantage of having a larger image-text dataset as we lack statistics for the number and types of concepts extracted. Moreover, CC3M is quite polished and small compared to other web-scale datasets, and having an analysis in this direction would shed some light also on the type of data pool we should consider to increase generalisability and (potentially) visual discriminability, i.e., I would expect different image-text datasets to have differences in their captioning style, and some could be more beneficial than others for the task. The authors could refer to [1] for a potential list of web-scale datasets to consider, or work with subsets of different dimensions of CC3M if the first approach is unfeasible.
- The authors should expand on the pool of concepts discovered by the model, i.e., describing their quantity w.r.t. other works, and providing some other statistics on the matter, e.g., average concept length, POS tags.

[1] Singh, Amanpreet, et al. "Flava: A foundational language and vision alignment model." CVPR. 2022.

**Strengths And Weaknesses:**

Strengths:
- The authors identified limitations of prior work (i.e., shortcuts and limited visual distinctiveness) and propose an approach that addresses both;
- The concepts learned with the CDL framework are more general and discriminative, and also align better with human judgement;
- The work includes a thorough evaluation on diverse benchmark datasets, and contains human evaluation to demonstrate the quality of the CDL concepts;
- By removing the two limitations identified, the authors developed a framework that discovers more interpretable concepts that should generalise better to multiple domains.

Weaknesses:
- The framework depends on multiple components, i.e., a VLM (CLIP), an LLM (GPT3), and a data pool (CC3M) and their replacement with other backbones or pools was not considered. Specifically, I believe that the rationale behind this work (i.e., more general concepts, more interpretable concepts) would gain from larger data pools, e.g., CC12M. It would be interesting to see how the framework performs with larger and smaller data pools;
- The final concept selection process relies on a generalisability score and a fixed budget for the number of concepts. I could see some benefits in having a data-driven method to address this point.

---

> ### Author Response · Authors · 2024-10-23
> **Response to the review**
>
> We appreciate the insightful feedback from the reviewer cs4X.
>
> ### **Weakness 1 & Requested Change 1: About the use of different data pools and larger datasets**
> Thank you for this valuable point! Our main contribution is the proposal of the CDL framework, which can be applied to any general-purpose image dataset to discover concepts and utilize these concepts in downstream image classification tasks. As our focus is on discovering common visual concepts shared by different objects and problem domains, as opposed to learning visual representations (e.g. CLIP visual encoder), we hypothesize that concept discovery can be performed in a more data-efficient manner.
>
> To validate this hypothesis, we conducted an ablation study to investigate the performance of our CDL framework on datasets of varying sizes (target datasets we perform object classification on, a random 1M subset from CC-3M, CC-3M, and CC-12M). From the results based on the interpretability metric, we observed that:
>
> 1. As expected, it is possible to discover visual concepts directly on target datasets (due to knowing the list of objects), which outperforms prior approaches significantly on interpretability.
> 2. Using 1M general purpose images for concept discovery already matches the "in-domain" concepts discovered from the target datasets.
> 3. Using more than 1M images only slightly improves the interpretability metric.
>
> These observations help us confirm the hypothesis that concept discovery can be performed on a moderate-sized, general-purpose image dataset.
> | Interpretability | ImageNet | Food-101 | CIFAR-100 | CIFAR-10 | CUB-200 | Flowers-102 | Stanford Cars | Aircrafts | Oxford Pets |
> |------------------|----------|----------|-----------|----------|---------|-------------|---------------|-----------|-------------|
> | LaBo             | 1.9      | 4.0      | 24.0      | 40.0     | 1.0     | 1.1         | 2.4           | 1.2       | 9.3         |
> | LM4CV            | 2.7      | 5.3      | 19.4      | 30.0     | 4.5     | 4.2         | 5.1           | 2.7       | 8.9         |
> | CDL with target dataset | 29.7     | 70.6     | 74.0      | 80.0     | 41.2    | 45.5        | 46.8          | 26.7      | 72.9        |
> | CDL with CC1M    | 29.5     | 72.9     | 77.4      | 80.0     | 43.5    | 49.0        | 49.1          | 28.2      | 76.4        |
> | CDL with CC3M    | 32.3     | 73.4     | 78.2      | 80.0     | 44.6    | 48.5        | 50.2          | 29.8      | 77.5        |
> | CDL with CC12M   | 32.7     | 73.6     | 78.8      | 90.0     | 44.9    | 49.0        | 51.4          | 30.6      | 77.2        |
>
> ### **Weakness 2: Data-driven parameter selection**
>
> Thank you for your thoughtful suggestions! For the quantity of concepts, the challenge lies in selecting a small set of high-quality, performant concepts. We followed the same strategy from LM4CV and chose 1x and 2x the number of categories as the budget for the concepts. This not only ensures fair comparison with prior work, but also accounts for the parameter count for the linear classifier. We do agree with the reviewer that certain target datasets may contain object categories that can be described by more diverse visual concepts, and an adaptable vocabulary size may be useful.
>
> For the generalizability score (alpha), there is a trade-off between usefulness and generalizability as we described in our submission: specific concepts (e.g., “white face with black eye patches” for “giant panda”) boost classification performance but have lower generalizability. In contrast, broader concepts (e.g., “four-legged animal”) are more generalizable but may reduce classification accuracy. We tuned alpha on the validation set to balance these factors, ensuring the concepts are generalizable without significantly impacting performance.
>
> ### **Requested Change 2: Expand on the pool of discovered concepts**
>
> **Quantity of Concepts**
>
> See  **Weakness 2**.
>
> **Average Concept Length**
>
> We measured the average concept length (the number of words in the concept) of baselines and our methods.
>
> |              | ImageNet | Food-101 | CIFAR-10 | CIFAR-100 | CUB-200 | Flowers | Stanford Cars | Aircrafts | Oxford Pets |
> |--------------|----------|----------|----------|-----------|---------|---------|---------------|-----------|-------------|
> | **LaBo**     | 5.7      | 5.7      | 4.9      | 5.3       | 5.5     | 6.0     | 5.4           | 4.9       | 3.8         |
> | **LM4CV**    | 5.1      | 4.8      | 4.8      | 5.0       | 4.3     | 5.4     | 5.0           | 4.4       | 3.2         |
> | **CDL**      | 4.8      | 4.9      | 4.4      | 4.6       | 4.2     | 5.2     | 4.8           | 4.1       | 3.1         |
>
> The results show that our CDL methods can discover more concise concepts.
>
> **Pos tags**
>
> As shown in Figure 6, our discovered concepts are descriptive terms such as “yellow bill with black tip” and “dark brown plumage”. About 90% of the discovered concepts consist of nouns or adjective-noun phrases.

---

### Review · Reviewer_yQ8g · 2024-11-14

**Summary Of Contributions:**

This work explores if pretrained vision-language models automatically learn visual concepts during their training, proposes a framework for extracting such visual concepts, and shows that such visual concepts can be used to train downstream concept bottleneck models. Prior works have focused on understanding if pretrained vision-language models automatically learn visual concepts, but fall short in two ways: first, their prompting strategies that probe for concepts are inconsistent, leading to differing conclusions about whether or not VLMs automatically learn visual concepts. Second, many prompting strategies might include non-visual concepts or include object categories in the prompts which can act as ‘shortcuts’, leading the VLM to rely on the object category rather than visual concepts during retrieval. To address this, this paper proposes a concept discovery and learning (CDL) framework. This framework first discovers general concepts using the CC-3M dataset, choosing those that have the most mutual information between a VLM and LLM. Then, concept bottleneck models can be trained on top of the discovered concepts.

**Audience:**

Yes

**Broader Impact Concerns:**

I have no concerns about the ethical implications of the work.

**Claims And Evidence:**

No

**Requested Changes:**

Overall, this work has potential. However, in its current form, many of its contributions are not clearly communicated and could be better highlighted with improvements in writing. Specifically, the organization of the paper feels more like a long list of ideas rather than presenting the main points of each section (e.g., contributions) followed by the specific details (e.g., how those contributions are implemented). This is particularly evident in two areas: (1) in the introduction, where many details of the method are included but lack the necessary background and context for clarity, and (2) in the experiments section, where the experiments are presented consecutively but each one lacks sufficient detail for understanding, and the specific research question being addressed is often unclear. Furthermore, I am unable to fully evaluate the utility of the proposed method, as it appears that some of the experiments (e.g., Figure 4) may present incorrect results.

**Strengths And Weaknesses:**

This work offers a valuable framework for automated concept discovery, with strong potential for application in downstream object detection tasks. The results are impressive; however, I believe the paper would benefit from a clearer presentation of the motivation and a more structured approach to experiments and results. My major and minor comments are detailed below.

Major Comments

Framing and introduction

- The motivation and proposed approach in this work is strong, but the contributions could be made more explicitly stated earlier in the paper. In its current form, it was not clear to me that the paper has three focuses: (1) exploring if pretrained vision-language models automatically learn visual concepts, (2) proposing a framework for extracting such visual concepts, and (3) training concept bottleneck models on the extracted concepts. Instead, the motivation (and title) highlight the idea of concept discovery, and minimize the contribution of learning. Thus, as I read through the manuscript, I was confused about the motivation behind the learning aspect of the framework and whether or not it was a contribution. Furthermore, this made it difficult to understand the motivation behind some of the experiments.
- The introduction includes method-specific details in paragraphs 4 and 5 that are difficult to follow without background and problem setup. It could be strengthened by staying high-level, and leaving specific details about the method to the methods section.
- The setup to Figure 1 can be further clarified. For example, it wasn’t clear if the experiment conducted image or text retrieval. Additionally, how are the shown concept prompts chosen (i.e., are they the top-k prompts, randomly selected, etc)?

Methods
- The methods section is challenging to follow due to the lack of a clear, cohesive description of the proposed approach. While aspects of the method are introduced briefly in the introduction and then described in various parts of the methods section, it’s difficult to grasp the exact steps of the entire process. Including an overview of the method that outlines each component in a concise manner would make the methods section much more accessible and easier to follow.
- Section 3.1 discusses the interpretation of weight matrix $W$. Typically, interpreting such weights as importance values relies on the assumption that the data for which W is trained on are standardized. Are the activation vectors used to train weight matrix W standardized? If not, one would need to calculate standardized regression weights [1] to interpret concept importance. Does the manuscript do this? If not, how does doing this affect the results? This is particularly important for evaluation around precision.
- In section 3.3, there are conflicting statements between the motivation of the paper and assumptions made about VLMs. The paper assumes “the vision-language pre-training learns powerful encoders for recognizing visual concepts.”, whereas the motivation behind the paper is understanding if pre-trained visual language models learn discoverable visual concepts. I think the assumptions about VLMs make sense, but this should be reflected appropriately in the motivation of the paper.
- In section 3.3, It was difficult to understand the reason for “realigning” the image-text interface, when the assumption is that the VLMs learn powerful encoders for recognizing visual concepts. Furthermore, I struggled to follow the methodology in this section. For example, why does one construct a concept-category association matrix $W_{llm}$ rather than learning the weight matrix $W$? Second, how is this related to the linear projection layers of the VLM (i.e., the architecture was not clear).
- In section 3.5, intervention accuracy is used to indicate high interpretability of visual recognition. However, it is unclear what the definition of intervention accuracy is and how it is calculated.
- In section 3.5, the paper describes precision as being measured with respect to each correctly classified image, where the concepts are ranked according to the dot product between concept activation vectors and the weight matrix. However, in section 3.1, the paper describes understanding the importance of each concept in terms of only the weight matrix $W$. Why choose the former (an instance-based explanation) for the evaluation rather than the later (a global explanation)?

Experimental setup
- For which concept bottleneck models was the concept association matrix learned as $W$ versus constructed as $W_{llm}$?
- Section 3.4 discusses choosing visual concepts for downstream benchmarks, but the benchmarks at this point had not been introduced. Thus, it was confusing to follow the experimental setup without knowing what the benchmarks actually were.
- Section 3.4 also briefly mentions generalizability, which is then redefined in section 3.5. This made it difficult to understand the analysis in 3.4, so it would be helpful to have the definitions come before.

Experiments

- Beginning in Section 4.2, I found it challenging to follow the various experiments and the specific research questions they address. To improve the flow of the paper, it would be helpful to clearly state the research question each experiment is designed to answer. As it stands, the current order makes it difficult to track the purpose of each experiment.
- The baselines in Table 1 lack sufficient explanation, making it unclear what CDL is being compared to—specifically, the CLIP and VDES baselines need further clarification. Additionally, Table 2 presents only a single value per column, although the main text suggests that this experiment was conducted across all datasets. Are the results in Table 2 averaged?
- It seems that Figure 4 may display incorrect results for this paper. Based on the experiment’s goal of comparing LaBo to CDL, the figure legend mentions “LaBo (Ours),” which is confusing. It would be helpful to clarify the comparison being made here and provide the correct experimental results.
- Figure 4 depicts a few-shot learning setting, but the experiment setup is unclear. In typical few-shot learning, a small number of training examples are used to fine-tune a classifier. However, Figure 4 mentions that "concepts are selected," which is ambiguous. It would be helpful to clarify what is meant by this.
- In Figure 5, it is unclear how many image examples and human annotations were used for these results. How were these examples selected, and how many annotations were made per example? Furthermore, how were such results calculated, and what is the Y-axis for these plots?
- In section 4.5, the paper performs an ablation study testing the utility of CC-3M versus concept discovery on downstream datasets. However, Table 5 only shows the evaluation in comparison to CUB on in-domain and ImageNet on cross-domain. Why were not all downstream datasets used in this ablation, and what was the reasoning behind choosing one dataset for each in- and cross-domain comparison?

Minor Comments

- Figure 1: VDES is not defined in the first column.
- Citation  missing for GPT-3-text-davinci-002.
- The text in the figures is very small and could be made larger for better readability

[1] 6Bring, J.: How to standardize regression coefficients. The American Statistician 48(3), 209–213 (1994)

---

> ### Author Response · Authors · 2024-11-27
> **Response to the review**
>
> We thank reviewer yQ8g for the detailed review and valuable suggestions. We have revised the paper in response to the feedback (the revision is red colored). Below, we address each of the questions and comments point by point.
>
> ## Frame and Introduction
> ### **Q1: Motivation of concept learning**
>
> Although visual concepts can be discovered from frozen pre-trained VLMs, they are not always perfectly aligned with images. By applying a self-supervised learning objective, we aim to further improve the quality of visual concepts by leveraging the knowledge already encoded in pre-trained VLMs and LLMs. We evaluate the concept quality before and after concept learning. The results are shown in Figure 5 in the revised paper. It shows that the quality of our discovered concepts outperformed the baselines, but after learning the quality is further boosted. In this work, we perform concept learning on specific object recognition datasets due to computational constraints. A potential direction would be to perform the concept learning on large, general datasets such as CC-3M.
> We revise the concept learning-related parts in the Introduction and Method sections to highlight its motivation.
>
> ### **Q2: Too many method details in introduction**
>
> We revise the paragraph 4 and 5 of the introduction, removing technical details and stressing the high-level motivations.
>
> ### **Q3: Set up for Figure 1 should be clarified**
>
> We clarify the set up of Figure 1 in its caption. The displayed concepts are those most closely aligned with the image according to CLIP image and text similarity.
>
> ## Methods
>
> ### **Q1: The method part is hard to follow**
>
> We reorganize the method part and add an overview with concise descriptions of each part at the beginning of the Method section.
>
> ### **Q2: Sec 3.1 - details of the weight matrix W**
>
> We clarify in Sec. 3.1 that the activations are standardized with z-score normalization.
>
> ### **Q3: Sec 3.3 - Motivation of the paper and the assumptions**
>
> In this paper, our assumption is if VLMs learn concepts, they can be extracted via the V-L interface (inspired by VDES [1] and other works). Hence, we aim to understand whether and to what extent visual concepts are encoded in pre-trained VLMs through concept discovery. We also aim to explore whether the discovered visual concepts can be further improved through self-supervised concept learning, without fine-tuning the pre-trained V-L backbones. We discuss how to properly validate the quality of the discovered and learned visual concepts via quantitative and human evaluation, to prove that VLMs do learn visual concepts.
>
> ### **Q4: Sec 3.3 - Methods of concept learning**
>
> We revise the concept learning part in Method and revise the concept learning figure (Figure 3) to clarify the framework of concept learning. As illustrated in Figure 3, we first obtain the concept activation through the linear projection layers of VLMs and then perform matrix multiplication $f(a) = a \cdot W_{LLM}$ to obtain the category prediction. We use the fixed $W_{LLM}$ instead of learning the weight matrix $W$ because we want to build a ground-truth concept-category association with LLM knowledge and thus focus on optimizing the image-concept alignment.
>
> ### **Q5: Sec. 3.5 - Details of interpretability**
>
> We utilize the “intervention accuracy” proposed by [2] to measure the interpretability of concepts. It measures how well the concept-category association matrix learned from VLM predicted concept activations aligns with human knowledge. We revise Sec. 3.5 to include more details of the calculation of intervention accuracy.
>
> ### **Q6: Sec 3.5 - Details of precision**
>
> The decision to use an instance-based explanation for evaluation rather than a global explanation stems from the inherent variability of visual concepts across images within the same category. The weight matrix $W$ provides a global explanation of the concept-category associations by indicating how much each concept contributes to recognizing a particular category on average. However, different images within the same category may contain varying visual concepts due to factors such as shooting angles. The precision evaluation would be inaccurate if we annotate whether the top-weighted concepts for a category in $W$ are present in all images of that category. To address this, we use the image-specific method described in Sec. 3.5, which tailors the concept ranking to each image, ensuring that the evaluation reflects the actual concepts present in that specific image.
>
> ## Citations
>
> [1] Sachit Menon and Carl Vondrick. Visual classification via description from large language models. In The Eleventh International Conference on Learning Representations, 2022.
>
> [2] Tian Yun, Usha Bhalla, Ellie Pavlick, and Chen Sun. Do vision-language pretrained models learn composable primitive concepts? Transactions on Machine Learning Research, 2023.

---

> ### Author Response · Authors · 2024-11-27
> **Response to the review**
>
> ## Experimental Setup
> ### **Q1: Whether CBMs are $W$ or $W_{LLM}$**
>
> We utilize $W_{LLM}$ as the ground-truth concept-category association during the concept learning. During the evaluation of the discovered and learned concepts, we train the concept bottleneck model $W$ from scratch to perform concept-based object recognition, and evaluate the learned CBM to measure the quality of concepts.
>
> ### **Q2: Should briefly introduce benchmark information in 3.4**
>
> We revise Sec. 3.4 to include detailed benchmark information.
>
> ### **Q3: The definition of generalizability is not consistent in 3.4 and 3.5**
>
> We revise Sec 3.4 and 3.5 to use the consistent definition of generalizability: A concept is generalizable if it can benefit the recognition of unseen objects. The generalizability score $G(c)$ in Sec. 3.4 is an estimate of the actual generalizability of a concept $c$.
>
> ## Experiments
> ### **Q1: Specify the research question each experiment addresses**
>
> We add an overview in the Experiment section to illustrate the research question each experiment addresses.
>
> ### **Q2: Baselines in Table 1 needs more clarification**
>
> We revise the presentation of Table 1 by focusing on the analysis of limitations of existing methods. Table 1 shows CLIP-based zero-shot classification performance with different prompting strategies. The comparison of Row 2 and Row 3 shows that even random concepts have minimal impact on zero-shot performance, while the comparison of Row 2 and Row 4 shows that the removal of category names results in a catastrophic decline in zero-shot performance. Table 1 conveys consistent information with Figure 1 that the concept-augmented text prompts do not offer conclusive evidence whether pre-trained VLMs learn to encode concepts because the category names usually serve as “shortcuts”.
>
> ### **Q3: Wrong information in Figure 4**
>
> We fix the typo in Figure 4. The legends should be “LaBo” and “Ours”.
>
> ### **Q4: Setup of few-shot experiments in Figure 4**
>
> In typical few-shot learning, a small number of training examples are used to fine-tune a classifier. In our method, the concept selection in Sec 3.4 also requires the examples from the training dataset. To keep the few-shot setting, we utilize the VLMs before concept learning and select the concepts with a set of a small number of training examples. We train the model with the same set of examples. We revise Sec 4.3 to clarify the setting.
>
> ### **Q5: Should add detailed settings for Figure 5**
>
> We revise Sec. 4.4 to include more details of human evaluation for the results in Figure 5. For human evaluation of “precision”, and “thoroughness”, we randomly select 100 images from each dataset and ask 3 human workers to annotate the “precision” and “thoroughness” of top-weighted concepts for each image as described in Sec. 3.5. More details are shown in Appendix Sec. D. The “precision”, and “thoroughness” of each dataset are calculated as the averages of these evaluations on the 100 images. The Y-axis of Figure 5 represents the percentage values for intervention accuracy, precision and thoroughness.
>
> ### **Q6: The dataset chosen strategy in section 4.5 (why only choose CUB)**
>
> Considering the cost of human evaluation for the “precision” and “thoroughness”, we select the CUB dataset as the downstream benchmark for the generalization evaluation instead of using all downstream datasets.
>
> ## Minor Suggestions
>
> ### **Figure 1: VDES is not defined in the first column**
>
> We add the definition of VDES in the caption of Figure 1.
>
> ### **Citation missing for GPT-3-text-davinci-002**
>
> We add the link to GPT-3-text-davinci-002.
>
> ### **The text in the figures is very small**
>
> We enlarge the text in the figures .
>
> ## Citations
>
> [1] Sachit Menon and Carl Vondrick. Visual classification via description from large language models. In The Eleventh International Conference on Learning Representations, 2022.
>
> [2] Tian Yun, Usha Bhalla, Ellie Pavlick, and Chen Sun. Do vision-language pretrained models learn composable primitive concepts? Transactions on Machine Learning Research, 2023.

---

### Review · Reviewer_4BJ9 · 2024-11-16

**Summary Of Contributions:**

This work introduces a Concept Discovery and Learning (CDL) framework, aiming to determine whether pre-trained vision-language models (VLMs) inherently capture visual concepts such as color, texture, and shape. By leveraging mutual information between visual and textual modalities, CDL identifies category-agnostic, visually discriminative concepts. The framework is demonstrated on 9 different datasets, showing its potential for interpretable and generalizable visual recognition (without the need for supervised fine-tuning).

**Audience:**

Yes

**Broader Impact Concerns:**

This paper proposes a general methodology, so I don't think it needs a broader impact statement.

**Claims And Evidence:**

Yes

**Requested Changes:**

As mentioned in the weaknesses above, the experiments have mostly focused on the classification task, but concept learning could also potentially benefit image segmentation performance. Additional experiments exploring this could benefit the scope of the paper.

**Strengths And Weaknesses:**

## Strengths
1. Using mutual information to uncover category-independent visual concepts is an interesting way to understand VLMs' internal representations.
2. Evaluation is done quite extensively on 9 diverse datasets. It's also good to see that human-evaluated metrics like precision and thoroughness are used - this improves the transparency in the VLMs' decision-making process(es).
3. The "self-supervised concept learning framework" is also interesting which enables CDL's scalability by aligning concepts to categories without additional annotations.
4. Performance-wise, CDL outperforms LaBo and LM4CV, which are SOTA in terms of accuracy and interoperability.

## Weaknesses
1. This point has also been mentioned in the Future Work section. I feel that this work focuses too much on individual concepts which has been explored significantly by prior work. What would have been more interesting is to see how VLMs capture spatial relationships. For example, techniques from scene graph generation, which model objects and their relationships, could be integrated into CDL.
2. Experiments mostly focused on the task of classification. But I wonder if concept learning can also benefit image segmentation where we are interested in localizing such concepts in an image?

---

> ### Author Response · Authors · 2024-11-27
> **Response to the review**
>
> We appreciate the constructive feedback from reviewer 4BJ9.
> ### **Weakness 1 - Lack of considering spatial relationships**
>
> We acknowledge that it is an important future direction to explore how pre-trained VLMs capture spatial relationships. We can leverage the proposed CDL framework as a tool to explore more properties of pre-trained VLMs such as concept localization. Our new results in Appendix Section H study how well VLMs encode visual concepts at object level, which is a first step towards modeling spatial relationships.
>
> ### **Weakness 2 & Request Changes - Application to localization tasks**
>
> We explore the benefit of visual concepts on the object detection task by localizing those concepts in images. We report the experimental details and the results in Appendix Section H of the revised paper. The results show that the concept discovery and learning can enhance the object detection performance of the existing framework.

---

### Decision · Action_Editor_qYTA · 2024-12-20

**Recommendation:** Accept as is

**Comment:**

After the revisions, all reviewers are in favor of accepting the paper. The AC is inclined to agree.

**Audience:**

Yes. The paper is of interest to a fairly large number of people in TMLR's audience.

**Claims And Evidence:**

Yes. All three reviewers agree that the submission's claims are supported by accurate, convincing, and clear evidence.